# Unveiling Myelodysplastic Syndromes: Exploring Pathogenic Mechanisms and Therapeutic Advances

**DOI:** 10.3390/cancers17030508

**Published:** 2025-02-03

**Authors:** Nishanth Thalambedu, Bhavesh Mohan Lal, Brent Harbaugh, Daisy V. Alapat, Mamatha Gaddam, Cesar Giancarlo Gentille Sanchez, Muthu Kumaran, Ankur Varma

**Affiliations:** 1Department of Hematology and Oncology, University of Arkansas for Medical Sciences, Little Rock, AR 72205, USA; nthalambedu@uams.edu (N.T.); bmohanlal@uams.edu (B.M.L.); mvkumaran@uams.edu (M.K.); 2Department of Pathology, University of Arkansas for Medical Sciences, Little Rock, AR 72205, USA; bharbaugh@uams.edu (B.H.); dvalapat@uams.edu (D.V.A.); 3Winthrop P. Rockefeller Cancer Institute, University of Arkansas for Medical Sciences, 4301 W. Markham Street, Slot # 508, Little Rock, AR 72205, USA; mgaddam@uams.edu (M.G.); cgentille@uams.edu (C.G.G.S.)

**Keywords:** myelodysplastic syndrome, acute myeloid leukemia, pathogenesis, classification, bone marrow dysplasia

## Abstract

Myelodysplastic syndromes (MDS) are a group of blood disorders which affect the bone marrow to produce healthy blood cells, leading to low blood counts and a risk of progressing to acute leukemia. The presentation of MDS varies widely among patients because it’s caused by different genetic changes. Recent advances in genetics improved our understanding of MDS, including how it develops and how to classify its subtypes. New classification systems, like those from the World Health Organization (WHO) and the International Consensus Classification (ICC), help physicians better diagnose and categorize MDS. Treatments are also advancing, with a focus on precision medicine by targeting the specific genetic causes of the disease in each patient to improve outcomes. This review explains how MDS develops, explores the updated classification systems, and discusses the latest treatment options, offering a clearer picture of this complex condition for better patient care.

## 1. Introduction

Myelodysplastic syndromes (MDSs) are a complex group of clonal hematological disorders marked by ineffective hematopoiesis and peripheral blood cytopenias, with an increased risk of progression to acute myeloid leukemia (AML) [1,2]. This heterogeneity in clinical presentation reflects the diverse genetic abnormalities and pathophysiological mechanisms underlying the disease [3]. Historically, MDS classification relied heavily on morphological criteria and clinical features; however, recent advances in molecular biology have significantly reshaped our understanding of these disorders [4].

Emerging genomic and molecular insights have revealed a broad spectrum of genetic mutations and chromosomal alterations that drive MDS pathogenesis [5]. These findings have led to the development of updated classification systems, including those from the World Health Organization (WHO) and the International Consensus Classification (ICC), which integrate genetic, molecular, and clinical data to better categorize MDS subtypes and predict disease outcomes [6,7].

The evolving landscape of MDS research also underscores the shift toward personalized treatment approaches [8]. Advances in targeted therapies aim to address the specific genetic and molecular abnormalities present in each patient’s MDS, moving beyond traditional treatment paradigms [9]. These developments offer the potential for more effective and individualized therapeutic strategies, improving patient outcomes and quality of life [10].

This review explores the intricate pathogenesis of MDS, highlighting the recent advancements in our understanding of its genetic and molecular basis. We will also examine the latest updates in MDS classification systems and discuss emerging therapeutic approaches that reflect the ongoing evolution in the management of this challenging group of disorders.

## 2. Pathogenesis of Myelodysplastic Syndromes

MDS is characterized by dysplasia in the myeloid lineage cells. The development of dysplasia in myeloid stem cells primarily involves two factors: the acquisition of somatic driver mutations and a dysfunctional immune system that fails to eliminate the abnormal stem cells [11].

### 2.1. Acquisition of Somatic Driver Mutations

In the majority of MDS cases, no specific cause is identified. However, DNA-damaging events from therapeutic exposure and non-therapeutic toxic substance exposure leading to MDS were noted in a minority of MDS cases. Therapeutic drugs commonly associated with dysplasia include chemotherapy agents such as antimetabolites, alkylating agents, topoisomerase II inhibitors, and radiation. The latency period for these agents varies, ranging from 2 years for topoisomerase II inhibitors to 6.5 years for alkylating agents and radiation [12]. Non-therapeutic toxic substance exposures—such as those experienced by atomic bomb survivors, individuals exposed to chemicals in agriculture and textiles, and veterans exposed to Agent Orange—also have variable latency periods [13,14].

Earlier studies among MDS patients revealed cytogenetic abnormalities causing chromosomal instability in regard to deletions and gains, which were thought to be the drivers of MDS, with the most commonly noted regions including del(7q) and del(5q) [15]. Other karyotypic abnormalities noted among MDS patients include del 17p, trisomy 8, del (20q), and complex karyotype, which is defined as the presence of ≥3 independent cytogenetic abnormalities [16].

Recent advancements in next-generation sequencing (NGS) have been utilized to gain a deeper insight into the pathogenesis of MDS. This has uncovered recurrent somatic mutations in key genes involved in hematopoiesis, such as GATA1, HOXA9, and KLF1, as well as in genes that regulate the expression of these primary genes by affecting various cellular processes including epigenetic regulation, defective splicing machinery, cellular metabolic adjustments, histone modifications, and signal transduction [17].

### 2.2. Common Gene Mutations in MDS Organized by Pathways

MDSs are driven by diverse genetic mutations disrupting key cellular pathways involved in hematopoiesis, epigenetic regulation, RNA splicing, and genome stability.

#### 2.2.1. Epigenetic Regulation

Mutations in genes such as DNMT3A and TET2 are frequently observed in MDS and contribute to abnormal DNA methylation patterns. DNMT3A mutations enhance protein dimerization, leading to hypermethylation of CpG islands, and are associated with poorer survival outcomes [18]. Similarly, TET2 mutations result in altered ten-eleven translocation 2 (TET2) dioxygenase activity, disrupting DNA methylation regulation [19,20]. Mutations in IDH1/2 also affect epigenetic regulation by producing 2-hydroxyglutarate (2-HG), an oncometabolite that inhibits enzymes involved in DNA and histone modifications, ultimately promoting oncogenesis [21,22,23].

#### 2.2.2. RNA Splicing Machinery

Mutations in RNA splicing genes, including *SF3B1*, SRSF2, U2AF1, ZRSR2, and LUC7L2, are highly prevalent in MDS and critical to its disease pathophysiology [24]. These mutations disrupt normal spliceosome function, leading to defective RNA splicing. Notably, *SF3B1* mutations are associated with MDS with ringed sideroblasts (MDS-RS), while SRSF2 mutations are linked to chronic myelomonocytic leukemia (CMML) [25,26].

#### 2.2.3. Transcriptional Regulation and Genome Surveillance

Bi-allelic mutations in *TP53*, a critical tumor suppressor gene, are associated with adverse outcomes in MDS [27]. The normal *TP53* protein regulates the transcription of genes essential for maintaining genomic integrity and is often referred to as the “guardian of the genome”. Loss-of-function mutations in *TP53* result in disordered proliferation and differentiation, contributing to oncogenesis [28].

#### 2.2.4. Genome Stability and Chromosomal Integrity

Cohesin complex mutations, including STAG2, RAD21, SMC1A, and SMC3, are implicated in MDS pathogenesis. These mutations affect the ability of cohesin proteins to regulate genome architecture dynamically, leading to defects in stem and progenitor cell proliferation and differentiation [29]. Cohesin subunit mutations exhibit unique biological characteristics and are associated with MDS and acute myeloid leukemia (AML) [30,31]. Cytogenetic abnormalities such as deletions (e.g., del(5q), del(7q), del(20q)) and trisomies (e.g., trisomy 8) are also commonly observed in MDS and contribute to chromosomal instability [15,16].

### 2.3. Dysfunctional Immune System

Another layer of MDS pathogenesis involves abnormal immune function, particularly in the bone marrow microenvironment. In up to 50% of patients with MDS, an increase in proinflammatory cytokines is observed due to the amplification of immune-related genes in myeloid stem and progenitor cells leading to abnormal differentiation and maturation of these cells, contributing to the development of MDS [32]. This finding aligns with the higher prevalence of MDS observed in individuals experiencing chronic immune stimulation [33].

Studies revealed there exists overexpression and downregulation of proteins from mutated genes involved in the innate immune system, and these are summarized in Table 1 [11,17].

Clinical trials targeting the inhibition of interleukin 1 receptor-associated kinase 4 (IRAK4) with IRAK4 inhibitor emavuserinib, among MDS patients harboring spliceosome mutations like U2AF1 and *SF3B1*, demonstrated a clinical response, and this inhibitor has also been shown to lower levels of IRAK4 among those who responded [34,35].

The Transforming Growth Factor β (TGF-β) pathway is crucial for maintaining hematological homeostasis by regulating the proliferation and differentiation of myeloid stem cells and their progenitors. This pathway involves a complex network influenced by various signaling cascades [35,36]. Among these, SMAD4 and SMAD3 are key players in the signal transduction from TGF-β receptors and represent potential therapeutic targets [37]. Luspatercept is a recombinant fusion protein that binds to TGF-β receptors, reducing signaling by SMAD2 and SMAD3, which promotes the maturation and differentiation of cells. The FDA has approved Luspatercept for treating anemia in low- to intermediate-risk MDS patients who are transfusion-dependent and have not responded to erythropoiesis-stimulating agents [38].

Another layer of pathogenesis of MDS involves amplified telomerase activity in clonal stem cells. It is coded by the TERT gene on chromosome 5p, and it is transiently active in normal myeloid stem cells to enable replication [39]. However, dysplastic stem cells exhibit high telomerase activity causing uncontrolled proliferation leading to ineffective erythropoiesis [40]. Imetelstat is a newly approved therapeutic option that binds and exerts its potent inhibitor effects against telomerase, thereby inducing apoptosis of malignant dysplastic cells and enabling the recovery of marrow hematopoiesis resulting in clinical outcomes [41].

## 3. Classification Systems

Classification systems for MDS have evolved significantly since its initial description as “pre-leukemia” in the French–American–British (FAB) classification, proposed by Dr. Bennett et al. in 1976 [42]. The first FAB classification and its subsequent revisions, including one in 1982, in which the term myelodysplastic syndrome was coined, as well as early classification systems from the WHO relied primarily on morphologic and clinical findings for diagnosis and subclassification of MDS [6,43]. As our knowledge of the genetic underpinnings of MDS and AML have evolved, morphologic features, such as the presence/absence of ring sideroblasts or the number of dysplastic hematopoietic lineages, have become less critical for subclassification to the point where the identification of a single gene rearrangement (e.g., *MECOM*, *MLL/KMT2A*) could warrant a diagnosis of AML rather than one of MDS. As accurate classification is crucial for prognosis and treatment planning, this section describes current updates to our classification systems, including the integration of genetic and molecular data, and the implications for clinical practice.

### 3.1. Current Classification Systems

Following the FAB classification, which formed the backbone of classification for roughly two decades, the WHO in 2001, began integrating the biological and clinical information available at that point in time to refine MDS classification. This classification system was published then subsequently revised in 2008, 2012, and 2016 [42].

Following the fourth edition of the WHO classification system in 2016, further integration of genetic, molecular, and clinical data informed many of the updates present in the fifth edition of the WHO classification system, first released in 2022 [43]. Concurrently, a separate international independent working group developed and published classification guidelines for hematopoietic neoplasms and coined them the International Consensus Classification (ICC) [7]. These classification systems/guidelines for diagnoses of myelodysplastic syndromes are discussed below and are compared in Table 2. Also in 2022, a new risk-scoring system, the Molecular IPSS (IPSS-M), was introduced, integrating clinical parameters, cytogenetic abnormalities, and somatic mutations in 31 genes, to stratify MDS patients into six risk categories [44].

The classification of MDS precursors including Clonal Hematopoiesis of Indeterminant Potential (CHIP), Clonal Cytopenia of Undetermined Significance (CCUS), and VEXAS (vacuoles, E1 enzyme, X-linked, autoinflammatory, and somatic) syndromes are handled similarly by the WHO fifth edition and ICC [7,43].

The fifth edition of the WHO simplifies the classification of MDSs into two broad categories: MDSs with defining genetic abnormalities and MDSs defined morphologically. In the former, there are three entities: (1) MDSs with low blasts and 5q deletion (MDS-5q); (2) MDSs with low blasts and *SF3B1* mutation (MDS- *SF3B1*); and (3) MDSs with biallelic *TP53* inactivation (MDS-bi*TP53*). The morphologically defined entities are MDSs with low blasts (MDS-LB); MDS, hypoplastic (hMDS); and MDSs with increased blasts (MDS-IB). In a change from the fourth edition, cases of MDS-LB (defined as <5% bone marrow blasts and <2% peripheral blood blasts) no longer require the distinctions of single-/multilineage dysplasia; however, the qualifiers of “with single lineage dysplasia”, “with multilineage dysplasia”, and “with ring sideroblasts” are retained as acceptable alternative terminologies (e.g., MDS-LB with ring sideroblasts) to improve the utility of the classification system in low-resource settings. MDS, hypoplastic is a distinct myelodysplastic neoplasm with low blasts and age-adjusted bone marrow hypocellularity, evaluated by a bone marrow core biopsy, with worse survival than patients with aplastic anemia but better outcomes than other types of MDS. Cases of MDS with increased blasts are subdivided into MDS-IB1 and MDS-IB2. Cases with 5–9% bone marrow blasts and/or 2–9% peripheral blood blasts are designated MDS-IB1, and with 10–19% blasts, they are designated MDS-IB2. The WHO also recognizes MDS with increased blasts and fibrosis (MDS-f) as a distinct subtype of MDS-IB when bone marrow blasts are between 5% and 20% and MF-2 (moderate) or MF-3 (severe) fibrosis is present. In the previous fourth edition of the WHO, MDS with ring sideroblasts (MDS-RS) was a distinct diagnostic entity requiring at least 5% ring sideroblasts in the presence of an *SF3B1* mutation or >15% ring sideroblasts when *SF3B1* is unmutated. However, the fifth edition eliminates the MDS-RS as a distinct entity, recommending cases of myelodysplastic syndrome with ring sideroblasts be classified as either MDS-LB or MDS-*SF3B1*, depending on mutation status (as noted above, “MDS-LB with ring sideroblasts” is retained as acceptable alternative terminology) [43].

The ICC retains some aspects of previous classification systems including morphologic classification of MDS with single lineage or multilineage dysplasia for cases of MDS without increased blasts. The ICC is less explicit than the WHO with the separation of genetically vs. morphologically defined MDS; nonetheless, the distinctions are present within the system. The genetically defined myelodysplastic syndromes in the ICC guidelines are several, similar to the WHO, and include the following: MDS with mutated *SF3B1* (MDS-*SF3B1*); MDS with del(5q)[MDS-del(5q)]; MDS, NOS without dysplasia (defined by the presence of at least one cytopenia and -7/del(7q) or complex cytogenetics and/or any MDS-associated mutations with a variant allele frequency of at least 10%, excluding multi-hit *TP53* or *SF3B1*); and MDS with mutated *TP53*. MDS cases with 5–9% bone marrow blasts and/or 2–9% peripheral blood blasts are designated MDS with excess blasts 1 (MDS-EB1), and cases of MDS with 10–19% blast are designated MDS/acute myeloid leukemia (MDS/AML). This last designation is in contrast to the WHO, which still requires blast counts of at least 20% for a diagnosis of acute myeloid leukemia (AML) in the absence of certain specific AML-defining genetic abnormalities (e.g., *PML::RARA* fusion, *RUNX1::RUNX1T1* fusion, etc). Like the fifth edition of the WHO, the ICC guidelines also eliminate MDS-RS as a distinct entity, and cases with dysplasia and ring sideroblasts are classified as either MDS-*SF3B1* or MDS-NOS, depending on their mutation status [7,43].

### 3.2. Implications for Clinical Practice

Perhaps the difference between the ICC and WHO with the most profound impact on management is MDS with 10–19% blasts. In the ICC, it is treated in lines of AML due to presumed poor outcomes, while in the WHO, it is treated in lines of MDS. Clinicians should keep this in mind and consider other patient- and disease-related factors prior to starting treatment.

MDS with increased blasts and fibrosis is a specific subtype of MDS-IB1 introduced by the WHO, which is not recognized by the ICC. Bone marrow fibrosis in prior studies was shown to be associated with *TP53*, *JAK2*, and *SETBP1* mutations and also correlated with higher white blood cell and blast cell counts with pronounced dyspoiesis in megakaryocytes, all leading to a poor prognostic factor [45]. But, currently, the role of fibrosis is not utilized as a risk factor in IPSS-R or IPSS-M models.

The genetically defined myelodysplastic syndrome in the ICC and WHO of MDS with *TP53*/MDS with bi-allelic *TP53* inactivation is diagnosed in the presence of *TP53* aberrations (either mutations or deletions) and is justified by its association with a poor prognosis with suboptimal outcomes even with allo-HSCT [46,47]. It is worth noting that identification of bi-allelic *TP53* inactivation requires either next-generation sequencing or *TP53* sequencing analysis in conjunction with a method of detecting copy loss (e.g., FISH for *TP53* locus at 17p13.1 or array comparative genomic hybridization).

Finally, in some instances, patients might qualify for a diagnosis of MDS in one system but not in the other. For example, the fifth edition of the WHO requires erythroid lineage dysplasia amongst the essential diagnostic criteria for MDS-*SF3B1*, while the ICC allows for MDS-*SF3B1* to be diagnosed in the absence of morphologic dysplasia. However, while such occurrences are a possibility, in reality, this mutation is almost always associated with dysplasia and RS to some extent [48].

## 4. Diagnosis and Management: Implications for Clinical Practice

### 4.1. Clinical Features

MDS is a disease of older adults, with more than 80% of patients being older than 60 years at diagnosis, and that affects males more than females, except for MDS with del(5q), which is more common in females [49,50]. The clinical presentation is variable, with a fraction of patients being asymptomatic at diagnosis and incidentally detected to have MDS on evaluation of abnormalities on routine complete blood count. Among the patients who are symptomatic, symptoms are usually due to the associated cytopenias.

Anemia is the most common cytopenia, and manifests as fatigue, exercise intolerance, light-headedness, shortness of breath, or angina [51]. Chronic fatigue is a very common symptom in patients with MDS and is often disproportionate to the degree of anemia [52]. Thrombocytopenia and neutropenia are less common and manifest with bleeding manifestations and recurrent infections, respectively. A total of 10–20% of patients with MDS have autoimmune manifestations in the form of vasculitis, connective tissue diseases, inflammatory arthritis, etc. The features of these autoimmune manifestations tend to be different from idiopathic autoimmune diseases, with more frequent unclassified and incomplete forms [53].

### 4.2. Evaluation and Diagnosis

One of the crucial steps in the evaluation of MDS includes the exclusion of alternative causes of cytopenias and/or dysplasia, such as folate or vitamin B12 deficiency, alcohol or drug use, congenital disorders, and infections (e.g., HIV); this step is essential, guided by clinical history and targeted investigations.

Anemia with hemoglobin levels < 10 g/dL is observed in 52% of MDS patients, often with normocytic or macrocytic red blood cells (RBCs) and anisocytosis [54,55]. Thrombocytopenia, defined as a platelet count < 100,000/mm^3^, is present in 40% of patients, although platelet morphology typically appears normal [54]. Neutropenia, with an absolute neutrophil count < 800/mm^3^, is less common, affecting fewer than 20% of cases, and may be accompanied by dysplastic neutrophils exhibiting pseudo-Pelger–Huet anomalies [56]. Peripheral blood blasts exceed 2% in nearly 50% of patients but remain below 20% by diagnostic criteria [54].

Bone marrow evaluation, including aspirate and biopsy, is critical for diagnosis. Morphologically, the bone marrow is often hypercellular with dysplasia across one or more lineages and <20% myeloblasts. Bone marrow fibrosis is reported in 12% of cases [54].

Cytogenetic abnormalities are identified in up to 50% of MDS cases, with del(5q), trisomy(8), del(20q), del(7q), and monosomy 7 being the most commonly observed [57]. Balanced translocations are uncommon. The presence of certain cytogenetic abnormalities such as t(8;21), inv(16), t(16;16), and t(15;17) excludes the diagnosis of MDS and suggests a diagnosis of AML [7].

Acquired somatic mutations are nearly ubiquitous in MDS and span a wide range of cellular processes. Frequently mutated genes include *SF3B1*, TET2, ASXL1, SRSF2, RUNX1, and *TP53*. The application of next-generation sequencing (NGS) is indispensable for detecting these mutations, as they have significant prognostic and therapeutic implications [58]. However, no single mutation is 100% pathognomonic of MDS.

The fifth edition of the World Health Organization’s classification of hematolymphoid tumors requires the presence of persistent cytopenias, with <20% blasts in peripheral smear/bone marrow, and either dysplastic morphology in ≥10% of cells in a given lineage or a characteristic cytogenetic/molecular feature to diagnose MDS [43]. Cytopenias are defined as hemoglobin (Hb) < 13 g/dL in males and <12g/dL in females for anemia, absolute neutrophil count < 1800/mm^3^ for leukopenia, and platelets < 150,000/mm^3^ for thrombocytopenia. No clear-cut time period is defined to identify persistent cytopenia. The defining cytogenetic/molecular features include *SF3B1* mutation, del(5q) (either alone or with one other abnormality other than monosomy 7 or del(7q)), and two or more *TP53* mutations, or one mutation with evidence of *TP53* copy number loss or copy neutral loss of heterozygosity [43].

### 4.3. Prognostic Factors

Various clinical, pathological, and molecular features that confer an adverse prognosis in a patient with MDS have been identified. Advanced age, the presence of comorbidities, and poor performance status all independently confer poor prognosis in a patient with MDS [59,60,61]. The presence of cytopenias and transfusion dependence are also poor prognostic factors [62,63]. As discussed previously, certain cytogenetics molecular abnormalities confer poor prognosis, including ASXL1, *TP53*, RUNX1, et cetera (65) [63,64]. Higher bone marrow blasts have consistently been associated with poor prognosis [63,65]. Using these prognostic factors, various risk stratification models have been created to categorize patients into different risk categories. IPSS-R and, more recently, IPSS-M are the suggested risk stratification scores to be used and are summarized in Table 3 [44,47]. These risk stratification tools may be used to divide patients into either having lower-risk MDS (LR-MDS) or higher-risk disease (HR-MDS).

## 5. Treatment of MDS

### 5.1. Overview of Treatment

The treatment strategy depends on various factors including risk stratification, the presence of symptoms/cytopenias, transfusion dependence, the presence of actionable mutations, the fitness of the patient, and patient preferences, among other things. Fitness as assessed by performance scores and comorbidity indices is more important than the chronological age of the patient in determining the treatment strategy.

### 5.2. LR-MDS

No treatment strategy has shown survival benefit in a randomized clinical trial compared with no treatment in patients with LR-MDS. The goal of treatment is to alleviate symptoms, improve quality of life, and reduce the dependence on transfusions. Treatment is reserved for patients with symptomatic cytopenias. Asymptomatic patients can be carefully observed, with adequate supportive care (Figure 1).

Anemia is the most common symptomatic cytopenia. Erythropoiesis-stimulating agents (ESAs) such as epoetin alfa and darbepoetin alfa have been used for anemia in MDS for many decades. In a randomized placebo-controlled phase 3 trial, darbepoetin alfa was compared with placebo in LR-MDS patients with Hb ≤ 10g/dL and serum erythropoietin (EPO) ≤ 500 mU/mL. Darbepoetin alfa significantly reduced transfusion incidence (36.1% versus 59.2% the in placebo group, *p* = 0.008) from weeks 5–24 [66]. In a meta-analysis, a standard dose of 30,000 to 40,000 U weekly was associated with an erythroid response rate of 49%, and a higher dose of 60,000 to 80,000 U weekly was associated with an erythroid response rate of 65%. ESAs’ impact on overall survival has not been consistently demonstrated [67,68]. In symptomatic patients with serum EPO < 500 mU/mL, we recommend at least a 3-month trial of ESAs.

In patients who do not respond to ESAs or those with serum EPO > 500 mU/mL, treatment strategy depends on various other factors. In patients with ring sideroblasts or *SF3B1* mutation, luspatercept can be used. In the phase 3 MEDALIST trial, involving patients with LR-MDS with ring sideroblasts and transfusion-dependent anemia, luspatercept use was associated with significantly higher rates of transfusion independence for 8 weeks or longer (38% vs. 13%, *p* < 0.001) [38]. In the interim analysis of the phase 3 COMMANDS trial, which compares upfront luspatercept in ESA-naïve LR-MDS patients with transfusion-dependent anemia with epoetin alfa, the luspatercept group was associated with a significantly higher rate of transfusion independence for at least 12 weeks, with a concurrent mean Hb increase of at least 1.5g/dL compared to epoetin alfa (59% vs. 31%, *p* < 0.0001) [69]. It is interesting to note that this benefit was in all LR-MDS patients and was not limited to patients with ring sideroblasts or *SF3B1* mutation.

In patients with del(5q) and anemia, lenalidomide has been shown to be effective. In a phase 3 trial involving 205 LR-MDS patients with del(5q) and transfusion-dependent anemia, the lenalidomide group had a higher rate of transfusion independence for ≥26 weeks compared to placebo (56.1% and 42.6% for lenalidomide 10 and 5 mg groups, respectively, versus 5.9% in placebo group, *p* < 0.001). Imetelstat can be used in patients with LR-MDS who are not responding to ESA or who have stopped responding to ESA. In the phase 3 IMerge trial involving LR-MDS patients with ESA-relapsed, ESA-refractory, or ESA-ineligible anemia, imetelstat was associated with a significantly higher rate of transfusion independence of at least 8 weeks (40% vs. 15%, *p* < 0.001) [41].

In patients with multilineage dysplasia, or patients with ESA-relapsed or ESA-refractory anemia, hypomethylating agents (HMAs), such as azacitidine and decitabine, can be used. Neither agent has been conclusively proven to be superior to the other [70]. Also, they have not been shown to prolong survival in patients with LR-MDS. While decitabine needs to be given intravenously, azacitidine can be given both intravenously and subcutaneously. An oral preparation of decitabine/cedazuridine is now available. Cedazuridine improves the oral bioavailability of decitabine by inhibiting cytidine-deaminase in the gut and liver [71]. It remains to be seen if this oral combination is as efficacious as intra-venous decitabine in patients with MDS.

### 5.3. HR-MDS

Allogeneic hematopoietic stem cell transplantation (allo-HCT) is the only “curative” therapeutic strategy in patients with MDS. The management of HR-MDS depends on whether the patient is transplant-eligible and if the donor is available. Where feasible, allo-HCT should be offered to the patient with HR-MDS (Table 4). In a multicenter biologic assignment trial, which compared outcomes of 50–75-year-old HR-MDS patients who received reduced-intensity allo-HCT with those who received HMAs or the best supportive care, the allo-HCT group had a significantly superior 3-year overall survival rate (47.9% vs. 26.6% in the HMA and supportive care group, *p* = 0.001). Importantly, the survival benefit was seen in all subgroups of patients. Among the patients who received allo-HCT, the estimated median disease-free survival was 26.1 months, and the median OS was not reached at a median follow-up of 28.4 months (IQR: 18.0–32.0 months) [72]. However, the outcomes of allo-HCT were not directly compared with HMAs (the HMAs and supportive care outcomes were clubbed).

In a prospective study comparing outcomes of HR-MDS patients who completed 4–6 cycles of azacitidine, and either received allo-HCT or continued azacitidine depending on the availability of suitable donors, patients in the allo-HCT group had a trend toward superior 3-year overall survival (50% vs. 32%, *p* = 0.12) and significantly superior event-free survival (34% vs. 0%, *p* < 0.0001). The cumulative incidence of transplant-related mortality at 1 year after allo-HCT was 19%. However, retrospective studies have shown a survival advantage with the use of allo-HCT over HMAs [72,73]. There is no randomized trial comparing the outcomes of upfront allo-HCT or bridging treatment followed by allo-HCT. But, in cases where the donor availability is expected to take time, patients can be started on bridging treatment while waiting for a donor.

Where feasible, a myeloablative conditioning regimen is preferred. Myeloablative conditioning has consistently been associated with fewer relapses and a slightly increased risk of transplant-related mortality over a reduced-intensity conditioning regimen. The differences in overall survival are not consistent [74,75,76]. There is no proven benefit of maintenance therapy after allo-HCT in patients with HR-MDS. In patients who are not transplant-eligible or where a suitable donor is not available, the treatment depends on the “fitness” of the patient, patient preferences, and availability of clinical trials. If the patient is fit, an intensive induction regimen along the lines of acute myeloid leukemia may be appropriate. In patients who are less fit, HMAs, HMA plus venetoclax, or the best supportive care are appropriate (Figure 2).

In a phase 3 trial involving 233 elderly HR-MDS patients (median age: 70 years), low-dose decitabine improved progression-free survival, but the overall survival and AML-free survival were not significantly different when compared to the best supportive care. Patients in the decitabine group had significantly reduced AML transformation rate at 1 year (22% vs. 33%, *p* = 0.036) [77]. Another retrospective study looking at outcomes with decitabine in HR-MDS patients found that decitabine was associated with an overall response rate of 70%, a median remission duration of 20 months, and a median survival of 22 months [78]. While there is no consistent survival benefit with decitabine compared to the best supportive care, azacitidine has demonstrated survival benefit in patients with HR-MDS. In a phase 3 trial comparing outcomes of azacitidine with conventional care in HR-MDS patients, patients in the azacitidine group had superior median overall survival (24.5 months vs. 15.0 months, *p* = 0.0001) [20].

### 5.4. Emerging Therapies in MDS

Bcl-2, an anti-apoptotic protein, plays an important role in the pathogenesis of MDS and is related to the progression of MDS and the transformation of MDS to AML. Venetoclax, a Bcl-2-inhibitor, has been tried in patients with MDS with good success. It functions as a BH3 mimetic, preventing BH3 proteins from binding to BCL-2. This disruption frees pro-apoptotic proteins BAK and BAX, causing mitochondrial outer membrane permeabilization (MOMP). As a result, cytochrome c is released into the cytoplasm, triggering the formation of the cytosolic apoptosome complex, caspase activation, and ultimately, cellular apoptosis [79]. In a study looking at the efficacy of a combination of azacitidine with venetoclax in treatment-naïve HR-MDS patients, the median overall survival was 26 months, and the complete response rate was 30% [80]. Another phase 1b trial evaluated venetoclax + azacitidine in 44 patients with relapsed/refractory HR- MDS after HMA failure, showing a median overall survival of 12.6 months. Marrow responses included CR (7%) and mCR (32%), with 36% achieving transfusion independence. Common Grade ≥ 3 adverse events included febrile neutropenia (34%) and thrombocytopenia (32%) [81]. A more recent phase 1/2 clinical trial looked at all oral regimens, including decitabine/cedazuridine with venetoclax in patients with HR-MDS or CMML. The combination was well-tolerated, and the overall response rate was 95% [82]. Randomized trials are needed to compare the efficacy of HMAs alone versus HMA plus venetoclax. The role of prophylactic venetoclax and azacitidine (Ven/Aza) maintenance therapy was evaluated in a phase 1 trial in 22 high-risk MDS/AML patients who underwent reduced-intensity allo-SCT with Ven/FluBu2 conditioning. The most common Grade 3–4 adverse events were transient leukopenia, neutropenia, and thrombocytopenia, with infections being rare (Grade 1–2). At a median follow-up of 25 months, the 2-year OS was 67%, PFS 59%, cumulative relapse 41%, and non-relapse mortality 0% [81,83].

CPX-351, an encapsulated form of cytarabine and daunorubicin, has been tried in HR-MDS. In a phase 2 trial involving 31 patients with HR-MDS or CMML, the overall response rate with CPX-351 was 87%, with a complete response rate of 52% [84]. The UK NCRI AML19 Trial compared CPX-351 with FLAG-Ida in younger adults with adverse-risk AML or high-risk MDS (around 30%). While overall survival and event-free survival were similar between groups, CPX-351 showed longer relapse-free survival (22.1 months vs. 8.35 months). Notably, patients with MDS-related gene mutations had better OS with CPX-351 (38.4 vs. 16.3 months), suggesting a potential benefit in specific subgroups [85]. Another single-center phase 1/2 study assessed the safety and efficacy of lower doses of CPX-351 in 25 patients with high-risk MDS(19) or CMML(6) following failure of hypomethylating agents. The overall response rate was 56%, with a median relapse-free survival of 9.2 months and a median OS of 8.7 months. Lower doses were well-tolerated, though cardiac toxicity occurred at the highest dose (75u/m2) in older patients [86].

Pevonedistat, a selective inhibitor of NEDD8-activating enzyme, when combined with azacitidine, led to significantly longer median event-free survival (20.2 months vs. 14.8 months, *p* = 0.045) and a trend toward superior median overall survival (23.9 months vs. 19.1 months, *p* = 0.24) when compared to azacitidine alone in patients with HR-MDS [87]. In the phase 3 PANTHER trial, pevonedistat plus azacitidine was compared with azacitidine monotherapy in 324 HR-MDS patients. The median event-free survival and median overall survival were not significantly different between the two groups. However, on post hoc analysis based on the number of cycles of the drugs, the median overall survival in receiving >3 cycles was 23.8 months in the pevonedistat plus azacitidine group as compared to 20.6 months in the azacitidine group (*p* = 0.021). For those receiving > 6 cycles, the median overall survival was 27.1 months versus 22.5 months in the azacitidine group (*p* = 0.008) [88]. A triplet combination showed encouraging results in a phase 1/2 study of the combination of pevonedistat, azacitidine, and venetoclax in older adults with secondary AML or MDS/CMML after the failure of hypomethylating agents. In the MDS/CMML cohort, the overall response rate was 75%, with 13% achieving CR and 50% achieving mCR. The most common Grade 3–4 adverse events were infection (35%), febrile neutropenia (25%), and hypophosphatemia (23%) [89].

In MDS patients with mutant IDH-1, ivosidenib, a mutant IDH-1 inhibitor has been tried recently. In a phase 1 study involving 19 relapsed/refractory MDS patients with mutant IDH-1 who failed standard-of-care therapy, once-daily ivosidenib resulted in a median overall survival of 35.7 months, with an overall response rate of 83% and a complete response rate of 39%. Also, 75% of RBC- and platelet transfusion-dependent patients became transfusion independent [23]. Olutasidenib, another selective mutated IDH-1 inhibitor, was recently tried in IDH-1-mutated AML and MDS patients in a phase 1/2 trial. A total of 65 patients had AML, and 13 patients had MDS. A total of six patients received olutasidenib monotherapy, and seven patients received olutasidenib plus azacitidine. In the monotherapy group, the overall response rate was 33%, with a complete response rate of 17%. In the combination therapy group, the overall response rate was 86%, with a complete response rate of 57% [90].

The IDH-2 gene is mutated in ~5% of patients with MDS. Enasidenib, an oral IDH-2 inhibitor, has been tried. In a study involving 50 MDS patients with mutated IDH-2, enasidenib was used as monotherapy in 23 patients with prior HMA therapy and in combination with azacitidine in 27 newly diagnosed IDH-2-mutated MDS patients. In the combination arm, the median overall survival was 26 months, and the overall response rate was 74%. In the monotherapy arm, the median overall survival was 20 months, with an overall response rate of 35%. Enasidenib was overall well-tolerated [91].

Programmed cell death protein 1 (PD-1) and PD-ligand 1 (PD-L1) expression is upregulated in patients with MDS, and it is further upregulated after exposure to HMAs. A combination of pembrolizumab, a monoclonal antibody targeting PD-1, and azacitidine was tried in patients with HR-MDS in a recent phase II study. In the 17 previously untreated patients, the overall response rate was 76%, with a complete response rate of 18%. In 20 patients with prior HMA failure, the overall response rate was 25%, and the complete response rate was 5%, with a median overall survival of 5.8 months [92].

In the phase 2/3 SWOG S1612 trial involving 37 patients with AML and 12 patients with MDS, patients were divided into two arms; one arm received azacitidine monotherapy, and the other arm received azacitidine plus nivolumab. There was a trend toward more early deaths in the azacitidine plus nivolumab group (24% vs. 4%), without a significant difference in the response rates. The trial was closed early due to the higher number of early deaths in the nivolumab group [93].

In a phase 1b trial, the GO29754 study, looking at the use of atezolizumab (PD-L1-inhibitor) alone or in combination in patients with MDS, was terminated before completion of recruitment due to unexpectedly high early death rates in HMA-naïve MDS patients receiving atezolizumab plus azacitidine, with no demonstrable efficacy [94]. In another phase 1/2 trial involving 30 patients with MDS who had relapsed after or were refractory to HMAs and 3 patients with CMML, a combination of atezolizumab and guadecitabine was tried. The overall response rate was 33%, with a complete response rate of 6%. Among patients with MDS, the median overall survival was 16.4 months. No dose-limiting toxicities were observed [95]. All the above-mentioned trials are summarized in Table 5.

## 6. Conclusions

Myelodysplastic syndromes (MDSs) are a highly heterogeneous group of hematological malignancies, with a complex pathogenesis involving somatic mutations, chromosomal abnormalities, and immune dysfunction. Recent advances in genetic and molecular research have significantly reshaped our understanding of MDS, highlighting the importance of integrating these insights into diagnostic and therapeutic strategies. The evolution of classification systems, particularly those from the WHO and ICC, underscores the shift from morphology-based diagnostics to a more precise, molecularly driven approach.

As MDS treatment moves toward personalized medicine, the development of targeted therapies holds great promise for improving patient outcomes, particularly in higher-risk subtypes. However, challenges remain, including the suboptimal response in certain genetic subgroups and the limited survival benefits of existing therapies for high-risk patients. Ongoing research into novel agents like IDH inhibitors, BCL-2 inhibitors, and immune checkpoint therapies offers hope for more effective treatments in the near future.

## Figures and Tables

**Figure 1 cancers-17-00508-f001:**
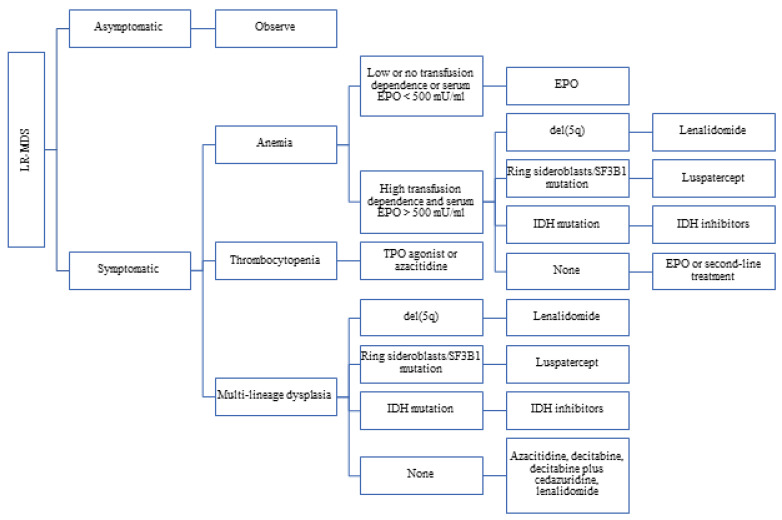
Approach to treatment of a patient with lower-risk MDS.

**Figure 2 cancers-17-00508-f002:**
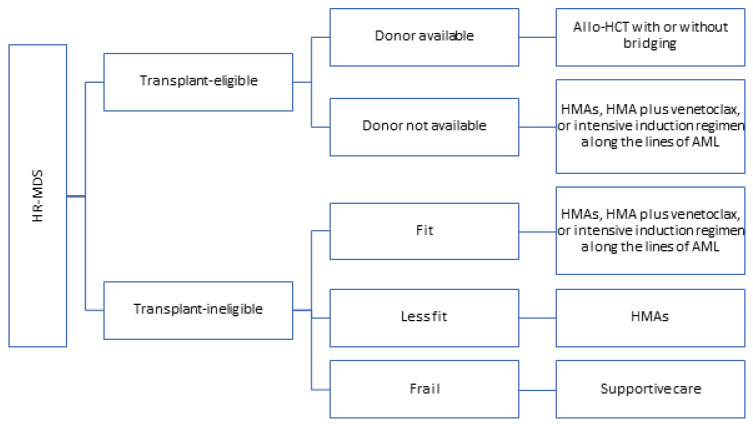
Approach to treatment of a patient with higher-risk MDS.

**Table 1 cancers-17-00508-t001:** Summary of Overexpressed and Downregulated Proteins from Mutated Genes Involved in the Innate Immune System in Myelodysplastic Syndromes (MDS).

Overexpressed	Downregulated
TLR IL1RAP CD14 TIRAP MYD88 IRAK4 IRAK1 TRAF6	miR-145 mIR-146a TIFAB

**Table 2 cancers-17-00508-t002:** Comparative WHO fifth edition and ICC classification for MDS.

WHO Fifth Edition	ICC
CHIP	CHIP
CCUS	CCUS
MDS with LB and *SF3B1* mutation	MDS with mutated *SF3B1*
MDS with LB and RS (acceptable alternative terminology)	Not included
MDS with LB and isolated 5q deletion	MDS with del(5q)
MDS with LB	MDS-NOS with SLD
MDS with LB	MDS-NOS with MLD
Not include	MDS-NOS without dysplasia
MDS, hypoplastic	Not included
MDS with IB1	MDS with EB
MDS with IB2	MDS/AML
MDS with increased blasts and fibrosis	Not included
MDS with bi-allelic *TP53* inactivation	MDS and MDS/AML with mutated *TP53*

**Table 3 cancers-17-00508-t003:** Comparison of IPSS, IPSS-R, and IPSS-M classification systems for myelodysplastic syndromes (MDSs).

Classification	IPSS	IPSS-R	IPSS-M
Key Parameters	- Cytogenetics (Good, Intermediate, Poor) - Bone marrow blast percentage - Hemoglobin - Platelet count	- Cytogenetics (refined into 5 risk groups) - Bone marrow blast percentage - Hemoglobin - Platelet count - Absolute neutrophil count (ANC)	- Molecular mutations (31 genes) - Cytogenetics (integrated with molecular findings) - Bone marrow blast percentage - Hemoglobin, platelet, and ANC counts
Cytogenetic Risk Categories	3 risk groups (Good, Intermediate, Poor)	5 risk groups (Very good, Good, Intermediate, Poor, Very poor)	Integrated molecular and cytogenetic risk profiles
Blast Thresholds	<5%, 5–10%, 11–20%, and 21–30%	<2%, 2–5%, 5–10%, >10%	Similar to IPSS-R
Prognostic Groups	- Low-risk: 0–1 - Intermediate-1: 1.5–2 - Intermediate-2: 2.5–3.5 - High risk: ≥4	- Very low: ≤1.5 - Low: >1.5–3 - Intermediate: >3–4.5 - High: >4.5–6 - Very high: >6	- Very Low: ≤1.5 - Low: >1.5–3 - Moderate Low: >3–4 - Moderate–High: >4–6 - High: >6–8 - Very High: >8

**Table 4 cancers-17-00508-t004:** Indications for hematopoietic stem cell transplantation (HSCT) in high-risk myelodysplastic syndromes (MDSs).

Criteria	Indications
Risk Stratification	- IPSS-R or IPSS-M: High or very high risk scores.
Disease Characteristics	- High blast count: >10% blasts in bone marrow or >5% blasts in peripheral blood. - Cytogenetic abnormalities: complex karyotypes (e.g., monosomy 7, del(5q), del(7q)) or high-risk chromosomal changes. - Failure of hypomethylating agents (azacitidine/decitabine).
Age and Performance Status	Younger age (60–70 years) and ECOG performance status 0–1.

**Table 5 cancers-17-00508-t005:** Summary of recent clinical trials in myelodysplastic syndrome therapies.

Treatment/Trial	Phase	Population	Results	Toxicities (Grade ≥ 3)
Venetoclax + Azacitidine in HR-MDS	1b/2	Treatment-naïve HR-MDS patients	Median follow-up of 31.9 months, 29.9% CR rate, with median OS of 26 months	Neutropenia (48.6%), thrombocytopenia (43%), febrile neutropenia (42.1%), anemia (34.6%), and infections (57%)
Venetoclax + Azacitidine in Relapsed/Refractory HR-MDS	1b	Relapsed/refractory HR-MDS after HMA failure	Median follow-up of 21.2 months. Median OS 12.6 months, 7% CR, 32% mCR	Febrile neutropenia (34%), thrombocytopenia (32%)
Decitabine-Cedazuridine + Venetoclax	1/2	Treatment-naïve HR-MDS and CMML	Median follow-up of 10.8 months, 95% ORR, and 49% proceeded to transplant	Thrombocytopenia (85%), neutropenia (74%), and febrile neutropenia (21%)
CPX-351 in HR-MDS/CMML	2	Treatment-naïve HR-MDS or CMML	Median follow-up of 16.1 months, 87% ORR, and 94% proceeded to transplant	Pulmonary (26%) and cardiovascular (19%)
CPX-351 vs. FLAG-Ida in HR-MDS (UK NCRI AML19 Trial)	3	Younger adults with high-risk MDS or adverse cytogenetic AML	ORR: 64% and 76%. Median OS: 13.3 vs. 11.4. Median RFS: 22.1 vs. 8.35	Non-hematological toxicities: 18% vs. 21%
CPX-351	1/2	HR-MDS or CMML after failure of HMA	ORR: 56%. Median OS: 8.7 months. Median RFS: 9.2 months	Febrile neutropenia (48%) and lung infection (20%)
Pevonedistat + Azacitidine vs. Azacitidine in HR-MDS/CMML/Low blast AML	2	Treatment-naïve HR-MDS or CMML or low blast AML	Median fu: 21.4 and 19 months. ORR: 70.9% and 60.4%. Median OS: 21.8 vs. 19 m. Median RFS: 21 vs. 16.6m	Neutropenia (33% vs. 27%), febrile neutropenia (26% vs. 29%), anemia (19% vs. 27%), and thrombocytopenia (19% vs. 23%)
Pevonedistat + Azacitidine vs. Azacitidine	3	Treatment-naïve HR-MDS or CMML or AML	Median EFS in the HR MDS cohort was 19.2 vs. 15.6 months. Median OS was 21.6 vs. 17.5 months	Anemia (33% vs. 34%), neutropenia (31% vs. 33%), and thrombocytopenia (30% vs. 30%)
Pevonedistat + Azacitidine + Venetoclax	1/2	HR-MDS or CMML or secondary AML after failure of HMA	In MDS/CMML cohort, ORR: 75%, CR: 13%	Infection (35%), febrile neutropenia (25%), hypophosphatemia (23%)
Ivosidenib in IDH1-Mutant MDS	1	Relapsed/refractory MDS with mutant IDH-1	ORR: 83% CR: 39% Median OS: 35.7m	Grade 1 QT interval prolongation (5.3%) and Grade 2 differentiation syndrome (10.5%)
Olutasidenib in IDH-1 Mutant AML/MDS	1/2	Monotherapy: 6 patients. Combination with Azacitidine: 7 patients	Monotherapy: ORR: 33%, CR: 17%. Combination with Azacitidine ORR: 86%, CR: 57%	Monotherapy: thrombocytopenia (28%), febrile neutropenia (22%), and anemia (22%) Combination therapy: thrombocytopenia (19%), febrile neutropenia (13%), neutropenia (13%), and anemia (20%)
Enasidenib in IDH-2-mutated MDS patients	1	Monotherapy: 23 patients with prior HMA failure. Combination with Azacitidine: 27 patients newly diagnosed	Monotherapy: ORR: 35%, mOS: 20 m. Combination with Azacitidine ORR: 74%, mOS: 26 m	Neutropenia (40%), nausea (36%), constipation (32%), and fatigue (26%)
Pembrolizumab + Azacitidine in HR-MDS	2	Untreated HR-MDS: 17 patients. Prior HMA failure: 20 patients	Untreated: ORR:76% CR: 18% mOS: NR Prior HMA: ORR:25% CR: 5% mOS:5.8 m	Pneumonia (32%), arthralgias (24%), and constipation (24%)
Nivolumab in HR-MDS/AML	2/3	12 MDS patients. Azacitidine (6) vs. Nivolumab + Azacitidine (6)	mOS: 6.9m vs. 5.2 m	Early deaths were higher in the combination group (24% vs. 4%), leading to early trial closure
Atezolizumab + Guadecitabine in R/R HR-MDS/CMML	1/2	30 MDS patients relapsed or refractory to HMA	ORR: 33% CR: 6% mOS: 16.4m	Deaths ≤ 30 days (9%); immune-related adverse events (IRAEs) occurred in 36% of patients

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
