# Peer review of "Unveiling Myelodysplastic Syndromes: Exploring Pathogenic Mechanisms and Therapeutic Advances"

_cancers, 2025, doi:10.3390/cancers17030508_

Round 1

Reviewer 1 Report

Comments and Suggestions for Authors

The review provides an overview of the pathogenic mechanisms and therapeutic advances in myelodysplastic syndromes (MDS). MDS are heterogeneous group of clonal hematological malignancies caused by multitude of genetic defects. Besides, the authors discus updated World Health Organization (WHO) and International Consensus Classification (ICC) which integrate genetic, molecular, and clinical data to better categorize MDS subtypes and predict disease outcomes. The paper systematically reviews literature reports and studies addressing pathogenesis of MDS, classification systems, diagnosis and management - implications for clinical practice, and treatment of MDS.

The review topic is completely covered. All the references are relevant. Namely, 40/92 are recent publications (within the last 5 years). It is of note that there are no self-citations which are expected from the authors of the Review. Table 1 is appropriate and understandable. However, the Table 2 titled "Comparative WHO 5th edition and ICC classification for MDS" is illegible. The manuscript is clear, comprehensive, relevant to the field and well structured. The conclusion is consistent with the evidence and arguments presented. Please accept my objections.

Author Response

Dear Reviewer,

Thank you very much for taking the time to review this manuscript. Please find the detailed responses below and the corresponding revisions/corrections in the re-submitted files.

Point-by-point response to Comments

Comment 1: However, the Table 2 titled "Comparative WHO 5th edition and ICC classification for MDS" is illegible.

Response: We corrected Table 2 to make it more legible.

Reviewer 2 Report

Comments and Suggestions for Authors

Review

Title

Unveiling Myelodysplastic Syndromes: Exploring Pathogenic Mechanisms and Therapeutic Advances

Authors

Nishanth Thalambedu , Bhavesh Mohan Lal , Brent Harbaugh , Daisy V. Alapat , Mamatha Gaddam, Cesar Giancarlo Gentille Sanchez , Muthu Kumaran , Ankur Varma *

Recent molecular advances in diagnosis and biopsy lead to novel and personalized treatment of myelodysplastic neoplasms. The authors have performed a comprehensive review of MDS pathogenesis and therapeutic interventions based on to the updated 5th edition of World Health Organization (WHO) and the latest International Consensus Classification (ICC) classifications.  The review has included a lot of historical and modern knowledge, and some results are useful and will be beneficial to clinicians, pharmacologists, and scientists. Please consider revision by reducing “traditional” part and expand treatment targets and mechanisms before publication.

Critiques

1). The session 1.  Please organize common gene mutations in MDS according to various pathways, including those targetable genes. Treatment should be discussion in the session 5.4.

2). The session 3.1 can be concise, omitting “historical classification”, and focusing on the comparison between the 5th ed WHO and ICC classification to emphasize the role of molecular diagnosis herein. For example, Morphology, e.g., presence or absence of RS, or one to multiple lineage dysplasia, become less critical in subclassification.; and identification of certain gene rearrangement could lead to change diagnosis from MDS to AML, e.g. MECOM, MLL/KMT2A.

The “3.3. Implications for Clinical Practice” should be emphasized in the review. The changes of classification lead to different therapeutic approaching

3). The session 4.2 please stays concise, while the prognostic factors are more important to determine treatment strategies. IPSS-M risk stratification scores should be included for details.

4). Traditional/conventional treatments are well illustrated in the review. However, novel studies, including clinical trials or potential individual medication appear too superficial. If it is possible, please summarizes it in a table that will be easy for readers to follow. Transplantation indications for MDS patients should be mentioned.

              Name of drug    mechanisms      indications, clinical trials              preliminary results

5). The session 5.4 is an important part.

Venetoclax, BCL-2 inhibitor, has been approved by FDA in 2018 for treatment of AML. Venetroclax + azacitidine, has been introduced as phase Ib study for high risk MDS. CPX-351/ Vyxeos for MDS has also bee explored. Please provide a bit detail in mechanisms, treatment effects and potential risks. Refer to the following recent publications: PMID: 36309981, PMID: 37422688, PMID: 38197938, PMID: 37171402 , PMID: 28013106, PMID: 37946611.

6). Please also refer to:Molecular Targeted Therapy and Immunotherapy for Myelodysplastic Syndrome. Int J Mol Sci. 2021 Sep 23;22(19):10232. doi: 10.3390/ijms221910232. Molecular Targeted Therapy and Immunotherapy for Myelodysplastic Syndrome. PMID: 34638574.

Minor critiques:

The words in the Table 1 should be centered to avoid the first letter is cut.

Author Response

Dear Reviewer, 

Thank you very much for taking the time to review this manuscript. Please find the detailed responses below and the corresponding revisions/corrections  in the re-submitted files. 

Point-by-point response to Comments 

Comment 1: The session 1. Please organize common gene mutations in MDS according to various pathways, including those targetable genes. Treatment should be discussion in the session 5.4  

Response: We agree with the comment. We have modified section 2.2 and organized it as per various gene mutation pathways in pages 2 & 3. We also removed the treatment part in section 2 as per your suggestion  

Comment 2: The session 3.1 can be concise, omitting “historical classification”, and focusing on the comparison between the 5th ed WHO and ICC classification to emphasize the role of molecular diagnosis herein. For example, Morphology, e.g., presence or absence of RS, or one to multiple lineage dysplasia, become less critical in subclassification.; and identification of certain gene rearrangement could lead to change diagnosis from MDS to AML, e.g. MECOM, MLL/KMT2A. 

The “3.3. Implications for Clinical Practice” should be emphasized in the review. The changes of classification lead to different therapeutic approaching 

Response: We agree with the comment. We have omitted the historical classification in section 3.1. We focused more on the comparison between WHO 5th ed and ICC classification in the new  section 3.1. The new section 3.2  includes the implication for clinical practice due to changes in classification.  

Comment 3: The session 4.2 please stays concise, while the prognostic factors are more important to determine treatment strategies. IPSS-M risk stratification scores should be included for details. 

Response: We agree with the comment. We have concised the evaluation and diagnosis part of MDS. We also added a new Table 3 in pages 8 and 9 comparing different prognostic scores. 

Comment 4: Traditional/conventional treatments are well illustrated in the review. However, novel studies, including clinical trials or potential individual medication appear too superficial. If it is possible, please summarize it in a table that will be easy for readers to follow. Transplantation indications for MDS patients should be mentioned. 

Name of drug mechanisms indications, clinical trials preliminary results 

Response: We agree with the comment. We have added a new table 4 for transplant indications on page 11. We also added another table 5 in pages 14-17 as per suggestion summarizing different trials in MDS. 

Comment 5: The session 5.4 is an important part. 

Venetoclax, BCL-2 inhibitor, has been approved by FDA in 2018 for treatment of AML. Venetroclax + azacitidine, has been introduced as phase Ib study for high risk MDS. CPX-351/ Vyxeos for MDS has also bee explored. Please provide a bit detail in mechanisms, treatment effects and potential risks. Refer to the following recent publications: PMID: 36309981, PMID: 37422688, PMID: 38197938, PMID: 37171402 , PMID: 28013106, PMID: 37946611. 

Comment 6: Please also refer to:Molecular Targeted Therapy and Immunotherapy for Myelodysplastic Syndrome. Int J Mol Sci. 2021 Sep 23;22(19):10232. doi: 10.3390/ijms221910232. Molecular Targeted Therapy and Immunotherapy for Myelodysplastic Syndrome. PMID: 34638574. 

Response: We agree with the comment. We have expanded section 5.4 and included information about recent trials as per your suggestions. We used all your references and included them in the manuscript in Section 5.4. We also added table 5 in pages 14-17 summarizing different trials in MDS. 

Minor critiques: 

The words in the Table 1 should be centered to avoid the first letter is cut. 

Response: We corrected the Table 1 to avoid the cut in the first letter